

# Comprehensive sentimental analysis of tweets towards COVID-19 in Pakistan: a study on governmental preventive measures

Muhammad Faisal Ali, Rabia Irfan and Tahira Anwar Lashari

Department of Computing, School of Electrical Engineering and Computer Science, National University of Sciences and Technology, Islamabad, Pakistan

## ABSTRACT

Sentiments are the key factors that lead to influence our behavior. Sentiment analysis is a technique that analyzes people's behaviors, attitudes, and emotions toward a service, product, topic, or event. Since 2020, no country has remained untouched by COVID-19, and the governing bodies of most countries have been applying several anti-pandemic countermeasures to combat it. In this regard, it becomes tremendously important to analyze people's sentiments when tackling infectious diseases similar to COVID-19. The countermeasures taken by any country to control the pandemic leave a direct and crucial impact on each sector of public life, and every individual reacts to them differently. It is necessary to consider these reactions to implement appropriate messaging and decisive policies. Pakistan has done enough to control this virus's spread like every other country. This research aims to perform a sentimental analysis on the famous microblogging social platform, Twitter, to get insights into public sentiments and the attitudes displayed towards the precautionary steps taken by the Government of Pakistan in the years 2020 and 2021. These steps or countermeasures include the closure of educational institutes, suspension of flight operations, lockdown of business activities, enforcement of several standard operating procedures (SOPs), and the commencement of the vaccination program. We implemented four approaches for the analysis, including the Valence Aware Dictionary and sEntiment Reasoner (VADER), TextBlob, Flair, and Bidirectional Encoder Representations from Transformers (BERT). The first two techniques are lexicon-based. Flair is a pre-trained embedding-based approach, whereas BERT is a transformer-based model. BERT was fine-tuned and trained on a labeled dataset, achieving a validation accuracy of 92%. We observed that the polarity score kept varying from month to month in both years for all countermeasures. This score was analyzed with real-time events occurring in the country, which helped understand the public's sentiment and led to the possible formation of a notable conclusion. All implemented approaches showed independent performances. However, we noticed from the classification results of both TextBlob and the fine-tuned BERT model that neutral sentiment was dominant in the data, followed by positive sentiment.

Corresponding author
Muhammad Faisal Ali,
mali.mscs18seecs@seecs.edu.pk

# INTRODUCTION

The COVID-19 pandemic, initially caused by SARS-CoV-2, also known as the coronavirus pandemic, has been spreading worldwide. The virus was first identified in a seafood marketplace in Wuhan, China, in late 2019 (*Velavan & Meyer, 2020*) and has infected half a billion people around the globe. Despite taking various steps to contain it, the virus spread dramatically, and the WHO declared this outbreak a pandemic on March 11, 2020. As of April 11, 2022, almost 500 million people have been affected, and 6 million precious lives have been lost worldwide during this pandemic. We can obtain the absolute and final statistical insights once the pandemic is over (*Hamzah et al., 2020*); however, it is one of the deadliest outbreaks in history. The coronavirus pandemic is not the first pandemic, and the world has faced different outbreaks and pandemics in different periods, such as SARS, MERS-Cov, and Ebola. The community has dealt with all such situations, and their efforts have stood fruitful (*Elder et al., 2020*). COVID-19 has affected every aspect of life and led to various social and economic repercussions worldwide. Society faced supply shortages, including food, medicine, and other necessities of life. Many businesses faced the worst situations and got affected by the measures taken by the authorities to combat this deadly pandemic. The misinformation or partial information through media, like the promotion of false propaganda, led a few people to panic buying and escalated tension in various other aspects (*Islam et al., 2020*). In this instability, no one can ignore the contribution and role of technology. It has helped the world in different and effective ways in making crucial and medically necessary decisions to contain the virus (*Soliman & Tabak, 2020*). The historical data has allowed researchers to define new boundaries for combating such fatal diseases on a large scale (*Pan et al., 2020*). Mass media and social media have been proven as the most significant data sources and contributors, more than ever before, in analyzing major trends. Along with other various uses, intriguingly, such data is also utilized for the sentiment and perception analysis of the public over time.

Sentiment analysis is instrumental in contributing to finding human behaviors present on social media applications. Various real-life domains such as business, health, tourism, education, and many more have utilized such analyses. Different recommender systems are performing very efficiently based on the outcomes from sentiment analysis. It can help analyze real-time trends and choose the right path per those trends (*Chaudhary & Naaz, 2017*). A very large-scale population can be monitored and tracked at a meager cost using sentiment analysis (*Choi et al., 2017*). Hence, it is a potent tool to help us see the community's general perception. In short, the role and contribution of computer technologies cannot be ignored when tackling deadly diseases or pandemics. These technologies have provided numerous opportunities to fight such outbreaks (*Goldschmidt, 2020*), especially determining the public's perception of social and mass media. Multiple research studies have shown that many outbreaks could have been contained more effectively and quickly if social media data had been considered (*Singh, Singh & Bhatia, 2018*). The COVID-19 pandemic has been a controversial topic on social media. Based on the recent events during the pandemic, sentiment analysis can help better understand the public's sentiment about these events and implement a policy that can keep up with the

public's perception. Past events have shown that whenever a pandemic or similar emergency has occurred, it has profoundly affected people (*Vahia, Jeste & Reynolds, 2020*). Such situations can be controlled, but there are many reasons why this process of overcoming may be delayed or slowed down; one is panic buying (*Islam et al., 2021*). It becomes alarming when the community feels fear and doesn't follow the precautions the concerned authorities give because people are either ill-informed or misinformed about crucial information. It just adds fuel to the situation (*Erku et al., 2021*). In such cases, it's imperative to study the perception of the general people from time to time so that the proper guidelines can be conveyed where needed.

Recently, when COVID-19 emerged in Wuhan, China, countries worldwide started to gear up to contain this deadly virus. Pakistan is a country that shares its border with China, so different preventive measures were introduced and implemented from time to time to combat the pandemic in the country. Pakistan has so far faced four dangerous waves of this virus. As of April 17, 2022, more than 1.5 million people have been affected, and over 30 thousand have died. The lives of ordinary citizens have been affected by the tough decisions to deal with these situations. The importance of considering public sentiment is discussed in detail; the motivation and objective behind this study are to analyze public sentiments for each of the countermeasures taken by the Government of Pakistan in containing the spread of the coronavirus. Some significant contributions of the underlying study are as follows:

- Five comprehensive datasets about different preventive measures taken to combat COVID-19 are collected.
- Four approaches are implemented on each dataset to analyze the sentiment about all preventive measures.
- The sentiment during four COVID-19 waves at different times in the country is analyzed and compared.
- The results obtained from the analysis can help concerned authorities in the future in conveying the proper guidelines and implementing decision policies following the sentimental tone of the public.

The article is organized as follows: "Related Work" presents a detailed overview of work related to the study under discussion. "Materials and Methods" provides the comprehensive characteristics of the dataset and the models' frameworks used during the research. The obtained results are presented in "Results," whereas "Discussion" contains a brief discussion about the critical findings of the underlying study. In the end, "Conclusion" concludes the presented case with some future directions.

## RELATED WORK

The research works under discussion below contain studies from various geographical zones and different implemented techniques, analyzing the data collected through multiple sources and for different periods.

*Liu & Liu (2021)* studied the public behavior toward the COVID-19 vaccine on Twitter data posted from November 1, 2020, to January 31, 2021. First, they used the VADER (*Hutto & Gilbert, 2014*) to determine the sentiment from the data and LDA (*Blei, Ng & Jordan, 2003*) for the topic modeling from positive and negative Tweets. They extracted five main topics for the positive sentiment (trial results, administration, life, information, and efficacy) and five for the negative category (trial results, conspiracy, trust, effectiveness, and administration). They also performed temporal analysis using the PELT algorithm (*Killick, Fearnhead & Eckley, 2012*). At last, they performed geographic analysis using the Twitter Geolocation package. This study was conducted when the vaccine was not available to the community. In another study, *Zhang et al. (2021)* extracted tweets between February 24, 2020, and October 14, 2020. They analyzed four different American cities and the same number of Canadian cities. This study determined how people are reacting to the coronavirus overall in different timespans and identified the public sentiment about three primary anti-epidemic measures "the lockdown," "mask," and "vaccine." They compared the results from the cities of both countries to each other. NCR (*Mohammad & Turney, 2013*) was used to classify each Tweet's emotions, and VADER to categorize the sentiments. They concluded that the number of infections with the coronavirus and the public sentiment was correlated.

*Surano, Porfiri & Rizzo (2021)* analyzed a Twitter dataset consisting of 1.3 million tweets from different regions of the USA from January 21 to May 31, 2020. This analysis was done on the containment measure "lockdown" and the change in perception and sentiment about it. They considered "education" and "wealth indicators" and collected the relevant data from US Census Bureau. For sentiment classification, this study used VADER and classified tweets into positive, negative, and neutral categories. Furthermore, two different time series were created for each region (state) and the DC area, with a daily fraction of positive and negative tweets. Multiple time series were collected and divided into three sections: before the occurrence of the pandemic, from its occurrence to the first peak, and from its rise to the end of May 2020. They studied each region's positive and negative portions and calculated the averages. In research on Singapore's "Circuit Breaker" *Ridhwan & Hargreaves (2021)* did a sentiment analysis on Twitter data from February 1, 2020, to August 31, 2020. They used VADER for the sentiment analysis, and tweets' compound score was considered for the classification. A pre-trained RNN was used to determine the emotions from the tweets. For topic modeling, LDA and GSDMM were used. They tried to find a correlation between the changes in these sentiments and the occurrence of real-life events in the country. Another study by *Dubey (2020)* performed sentiment analysis on the Twitter data extracted from 12 different countries from March 11, 2020, to March 31, 2020. This research used the NRC model, which classifies the text based on eight emotions defined by *Plutchik (1980)*, to categorize the sentiments of the tweets in the form of expressed emotion rather than the numerical value like in VADER. This study had two analysis phases; in the first phase, they discovered that Belgium was the city where 63% of the tweets showed positive sentiment and only 37% were negative. China was the country containing the most negative tweets from its population. In the second

phase, they performed the analysis with the emotions and observed that the USA, France, and China contained the highest no. of tweets with an "anger" tone.

*Samuel et al. (2020)* performed sentiment analysis on data posted on Twitter from February 2020 to March 2020, targeting the keyword "corona." They tried identifying the public sentiments related to the outbreak using COVID-related tweets and R statistical software with sentiment analysis packages. This study was limited to the United States, where they showed insights into fear sentiment over time in the country as the pandemic was approaching its peak level. They performed a textual analysis aided by the visualization of descriptive text. Moreover, they used two machine learning classification models, the naïve Bayes classifier and logistic regression, and compared their performance on textual analysis of tweets of different lengths. The naïve Bayes method performed well on short tweets with an accuracy of 91%, while logistic regression gave a reasonable classification accuracy of 74%. Both models performed relatively weakly for tweets containing longer text. In another study conducted in the Philippines for sentiment analysis about the COVID-19 vaccine, (*Villavicencio et al., 2021*) extracted tweets weekly from 1st to 31st March 2021. These extracted tweets were 11,974 in total and were both in English and Tagalog language. This was the first month of the vaccination drive in the country. This data was manually annotated with three different sentiments: positive, negative, and neutral. They used the TF-IDF approach for vectorization and the naïve Bayes classifier for classification. They evaluated the model with k-fold cross-validation with k = 10, which achieved an accuracy of 81.77%. (*Machuca, Gallardo & Toasa, 2021*) performed a sentiment analysis on Twitter data posted from January to July 2020. They extracted the data for every month with the keyword "#coronavirus" and studied using TF-IDF vectorizer, which assigns a weightage to a word within the text. They used a pre-trained binary logistic regression to classify the tweets. The applied model obtained an overall accuracy of 78.57%. They discovered that 46% of the tweets showed negative sentiments, and 54% were positive ones.

*Garcia & Berton (2021)* performed a topic detection and sentiment analysis on English and Portuguese tweets. They got the top ten topics under discussion and analyzed the data discussed on Twitter over time. To avoid the weak performance on short text with a reason for the lack of word co-occurrence information, they used GSDMM for topic modeling. At the same time, they combined three recent embedding models for feature extraction, *i.e.*, sBERT (*Reimers & Gurevych, 2019*), mUSE (*Yang et al., 2019*), and FastText (*Bojanowski et al., 2017*). For English tweets, they used CrystalFeel (*Gupta & Yang, 2018*) for sentiment analysis. They also compared performance in both languages in this study. *Singh, Jakhar & Pandey (2021)* performed a sentimental analysis on two different datasets, one containing tweets from India and the other containing data from worldwide. Their dataset contained tweets over 80 days, from January 20 to April 25, 2020. They used mRMR (*Radovic et al., 2017*) for feature selection and the BERT model (*Devlin et al., 2018*) to classify tweets. They achieved 93.89% validation accuracy and concluded that the people of India had a positive sentiment in those days compared to the rest of the world. In another study, *Kumar (2022)* performed a sentiment analysis on the public sentiment in India over time in multiple phases of lockdown and its gradual lifting. This study used the Sentiment140 dataset for

the training and a manually annotated tweets dataset for testing purposes. A hybrid deep learning approach that consisted of BiLSTM and CNN was used for the sentiment classification. The primary purpose of adding CNN along with the BiLSTM was feature extraction. This hybrid model showed an overall accuracy of 90%.

*Zhou et al. (2021)* conducted sentiment analysis research in Beijing, China. They collected Sina Weibo data from the initial days of the virus, from January 13 to February 28, 2020, and at the time of its re-emergence, from June 4 to July 20, 2020. They divided the data into subsets based on three countermeasures, lockdown, test-trace-isolate strategy, and the ban on the gatherings. The LDA was used for the topic extraction and Baidu for the categorizing sentiments. They observed that the people of China greatly supported the test-trace-isolate measure (60%) in the initial days of the pandemic. This support increased to the highest level (90%) during the re-emergence of Corona cases in the country. In a study conducted by *Tan et al. (2021)* on Sina Weibo, several public posts from different accounts were collected and analyzed across 31 Chinese provinces from January 2018 to December 2020. They followed the Zipf distribution to normalize the number of Weibo posts per account and used a professional Chinese sentiment API named Tencent NLP Product to determine the sentiment. In major, they focused on the economic indicators and analyzed the public sentiment before and during the pandemic. They discovered that the pandemic had had severe effects on public mental health.

Above, we have discussed multiple studies from various geographical regions analyzing data collected from different timeframes. It can be observed that there is no comprehensive research conducted in the region of Pakistan. Pakistan is a very close neighbor to China, where this deadly virus emerged, and has taken multiple steps to combat the pandemic. Although a few surveys have been done in Pakistan to check the public sentiment regarding COVID-19, all such surveys were done on a tiny scale where a range of 500–1,000 people participated. As the sentiment analysis and an overview of the public perception from social media, where the public has a significant presence, is essential to be extracted, which can help in implementing various needy steps according to the situation in the future, this study is performed with the mentioned purpose. An overview with salient points of studies discussed above is presented in Table 1.

## MATERIALS AND METHODS

Various studies conducted by different authors have adopted different steps in performing sentiment analysis; however, the central workflow remains common, which is depicted more thoroughly in Fig. 1.

First, a source platform is selected on which one wants to perform the analysis. For example, when it comes to social media analysis, a user can choose from various sources like Twitter, Instagram, Facebook, Reddit, *etc.* The selection of a platform depends upon the purpose, and it varies from analysis to analysis. After finalizing a platform, a user needs to extract the data. Every platform has its different ways of data gathering. Users can target one or more specific keywords to extract the information they want. The extracted data will also be in various forms, such as posts, tweets, news, or text. Third, comes data pre-processing, which includes multiple steps such as cleaning; the removal of stopwords,

**Table 1 An overview of studies discussed in related work.**

| Sr. # | Year | Data Source | Timespan | Region | Analysis approach |
|---|---|---|---|---|---|
| 1 | 2021 | Sina Weibo | Jan 18–Dec 20 | Mainland China | Tencent API |
| 2 | 2021 | Sina Weibo | Jan 20–Feb 20 | China | Baidu (OSS) |
| 3 | 2021 | Twitter | Jan 20–Apr 20 | India *vs* RoTW | Automated |
| 4 | 2022 | Twitter | Mar 20–onward | India | Automated |
| 5 | 2020 | Twitter | Feb 20–Mar 20 | USA | Automated |
| 6 | 2021 | Twitter | Nov 20–Jan 21 | USA (State & National Level) | Rule-based |
| 7 | 2021 | Twitter | Apr 20–Aug 20 | USA & Brazil | Automated |
| 8 | 2021 | Twitter | Feb 20–Oct 20 | USA & Canada | Rule-based |
| 9 | 2021 | Twitter | Jan 20–May 20 | USA | Rule-based |
| 10 | 2021 | Twitter | Feb 20–Aug 20 | Singapore | Rule-based |
| 11 | 2021 | Twitter | Mar 2021 | Philippines | Automated |
| 12 | 2020 | Twitter | Jan 20–Jul 20 | Multiple countries | Automated |
| 13 | 2020 | Twitter | Mar 2020 | Multiple countries | Rule-based |

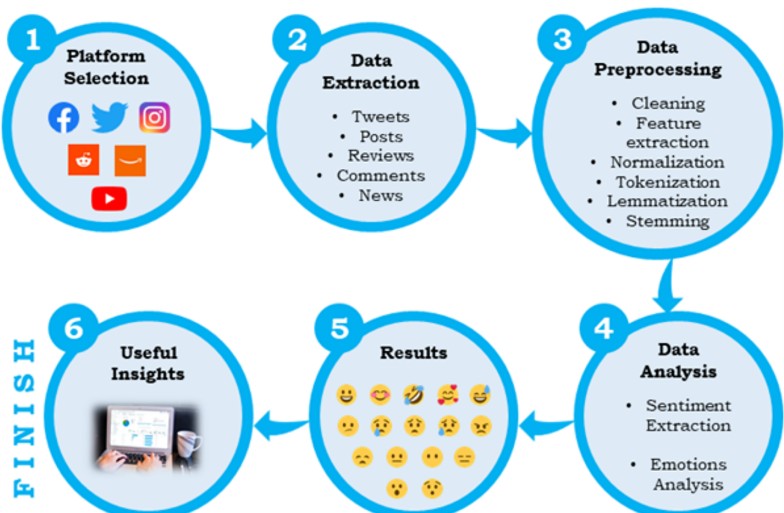

**Figure 1 Workflow of sentiment analysis or opinion mining.**

duplicates, and unwanted portions of text; feature extraction; structuring of the data and extraction of meaningful portions, normalization; converting all data to the same form such as lowercase, tokenization; converting the text to some keywords or tokens which contain some specific meanings, stemming; removal of last few characters of a word, and lemmatization; converting the text form to a meaningful base form. The pre-processed data is passed for sentiment analysis in the final phase, and some valuable findings are extracted.

## Datasets collection

The data was extracted from the social platform Twitter using the Snscrape library (github.com/JustAnotherArchivist/snscrape). Twitter has the edge over other social media platforms when analyzing a specific topic or a trend. It's the platform that networks ideas rather than people like Facebook. A post on Facebook may contain different topics because there is no such restriction on the text length, but on the other hand, Twitter allows only 280 characters per tweet. So, a tweet is usually relevant to a specific topic. Snscrape is an instrumental library that enables one to extract tweets without any personal API keys. It features extensive search options that permit customizable searches and can return thousands of records in minutes. Twitter has complex phenomena about fetching tweets by location. There are currently two ways to do so; either if a tweet is geotagged or if a site is mentioned in the user's profile. As per Twitter, a significantly less portion of tweets is geotagged, so it is not an optimal way. On the other side, many people have a location in their bio, but they can fill it up with whatever they want. Some are very kind and write properly, such as "Karachi, Pakistan," but a few write things like "My Dad's House." It was found that some Twitter algorithms work in a way in its advanced search where they can find tweets by location if it is a part of a user's bio and assume that all tweets coming from this profile will be tweeted from or around that specific location, or if the tweet is geotagged. There are two ways to make such an algorithm work for us; we can either provide coordinates of a location with a radius, and it will return with tweets posted in that radius, or we can use the "near" feature, such as "near: Karachi." As per our requirement, we needed the data from a country, *i.e.*, Pakistan. It was not an excellent way to provide coordinates of a country and a radius to collect the data, as it can result in inaccurate records. So, the best approach was to scrape tweets by cities and yield the best possible results. The data was extracted from the 100 most populated cities of Pakistan, presented in Table 2, targeting a radius of 50 km for each city.

Pakistan has adopted various preventive measures from time to time in fighting COVID-19 (*Jabeen et al., 2020*), including the closure of educational institutes, suspension of flight operations, lockdown of business activities, enforcement of several standard operating procedures (SOPs), and the commencement of the vaccination program. These were the steps that affected public life the most. Data were extracted for each of those countermeasures with the help of relevant keywords in specific periods, whose details are given in Table 3. These datasets are available for the community to use for further research (*Ali, 2022*).

## Data pre-processing

The process of data pre-processing is fundamental in data analysis. Here unwanted pieces of data are removed, and only a useful portion is kept. There can be several unwanted things in a tweet, like hashtags, mentioned user handles, *etc*. Table 4 represents the steps adopted while pre-processing the extracted raw text. It is pertinent to mention that Twitter hashtags can also contain some sentiments sometimes. In our case, most of them were related to COVID-19, such as #Coronavirus, #COVID, #Lockdown, #Vaccination, *etc*., and are present in almost every tweet. Such hashtags had no specific encoded sentiment

**Table 2 Most populated cities of Pakistan from which data was extracted.**

| Sr.# | City | Sr.# | City | Sr.# | City |
|---|---|---|---|---|---|
| 1 | Abbotabad | 34 | Islamabad | 67 | Mirpur Mathelo |
| 2 | Ahmedpur East | 35 | Jacobabad | 68 | Multan |
| 3 | Arif Wala | 36 | Jaranwala | 69 | Muridke |
| 4 | Attock | 37 | Jatoi | 70 | Muzaffarabad |
| 5 | Badin | 38 | Jhang | 71 | Muzaffargarh |
| 6 | Bahawalnagar | 39 | Jhelum | 72 | Narowal |
| 7 | Bahawalpur | 40 | Kabal | 73 | Nawabshah |
| 8 | Bhakkar | 41 | Kamalia | 74 | Nowshera |
| 9 | Bhalwal | 42 | Kamber Ali Khan | 75 | Okara |
| 10 | Burewala | 43 | Kāmoke | 76 | Pakpattan |
| 11 | Chakwal | 44 | Kandhkot | 77 | Peshawar |
| 12 | Chaman | 45 | Karachi | 78 | Quetta |
| 13 | Charsadda | 46 | Kasur | 79 | Rahim Yar Khan |
| 14 | Chiniot | 47 | Khairpur | 80 | Rawalpindi |
| 15 | Chishtian | 48 | Khanewal | 81 | Sadiqabad |
| 16 | Dadu | 49 | Khanpur | 82 | Sahiwal |
| 17 | Daharki | 50 | Khushab | 83 | Sambrial |
| 18 | Daska | 51 | Khuzdar | 84 | Samundri |
| 19 | Dera Ghazi Khan | 52 | Kohat | 85 | Sargodha |
| 20 | Dera Ismail Khan | 53 | Kot Abdul Malik | 86 | Shahdadkot |
| 21 | Faisalabad | 54 | Kot Addu | 87 | Sheikhupura |
| 22 | Ferozwala | 55 | Kotri | 88 | Shikarpur |
| 23 | Ghotki | 56 | Lahore | 89 | Sialkot |
| 24 | Gojra | 57 | Larkana | 90 | Sukkur |
| 25 | Gujranwala | 58 | Layyah | 91 | Swabi |
| 26 | Gujranwala Cantt. | 59 | Lodhran | 92 | Tando Adam |
| 27 | Gujrat | 60 | Mandi Bahauddin | 93 | Tando Allahyar |
| 28 | Gwadar | 61 | Mansehra | 94 | Tando Mohd. Khan |
| 29 | Hafizabad | 62 | Mardan | 95 | Taxila |
| 30 | Haroonabad | 63 | Mianwali | 96 | Turbat |
| 31 | Hasilpur | 64 | Mingora | 97 | Umerkot |
| 32 | Hub | 65 | Mirpur | 98 | Vehari |
| 33 | Hyderabad | 66 | Mirpur Khas | 99 | Wah Cantonment |
| 100 | Wazirabad | | | | |

which could add something to the overall sentiment of a tweet, so we removed them during pre-processing.

## Approaches selected for the analysis

There were four approaches implemented during the underlying study. The first two, VADER and TextBlob, were rule-based techniques. Third, a pre-trained embedding-based

**Table 3 An overview of datasets extracted against each countermeasure.**

| Countermeasures | No. of tweets extracted | Time period (Months) | Starting from (Month) | Extracted until (Month) |
|---|---|---|---|---|
| Closure of educational institutes | 672,465 | 19 | Mar 2020 | Sep 2021 |
| Suspension of flight operations | 159,418 | 22 | Mar 2020 | Dec 2021 |
| Lockdown of business activities | 646,679 | 22 | Mar 2020 | Dec 2021 |
| Standard operating procedures (SOPs) | 214,019 | 22 | Mar 2020 | Dec 2021 |
| Vaccination program | 162,548 | 15 | Dec 2020 | Feb 2022 |

**Table 4 Pre-processing steps taken during the analysis.**

| Sr. # | Pre-processing step | Description | Example |
|---|---|---|---|
| 1 | Hashtags | Hashtags help extract data relevant to a specific topic but are not so useful after getting data scraped. So, these hashtags were removed from the data. | #COVID-19, #Vaccination, #Lockdown |
| 2 | User handles | On Twitter, each user has a unique username, such usernames help specify a person but have no use in the text analysis. So, all user handles were removed. | @OfficialNcoc, @fslsltn, @WHO |
| 3 | URLs | A tweet may contain an external link or an attachment in the form of a hyperlink. All such things provide no benefit in text analysis and were removed. | bit.ly/3mcQZEs |
| 4 | Useless characters | A few useless characters were also removed from the text, just to make it neater and cleaner. | @, #, \|, \n, \t, \r |
| 5 | Multiple spaces | Multiple spaces occurred when the above pieces of texts were removed from the data, these were also removed to get data in a normal form. | More than one space |
| 6 | Null values | Null or empty records were also removed. | Any empty record |
| 7 | Duplicates | All duplicates amongst the records were discarded, so that each tweet should be unique and present a meaningful insight. | Any duplicate record |
| 8 | Punctuations | Some techniques do not require punctuations as they work with the tokens or lexicons only, so punctuations were removed while implementing such techniques. | ?, !, ;, :, -, … |
| 9 | Stopwords | Stopwords do not add much meaning in lexicon-based approaches. These were removed while applying such approaches. | My, we, it, to, from, be, has, have, do, so |

model, "Flair" (*Akbik, Blythe & Vollgraf, 2018*), was used. In the end, a fine-tuned transformer-based model, "BERT," was considered for sentiment classification. This fine-tuned BERT was first trained on annotated data and then used on the unseen data extracted from Twitter discussed above.

## VADER

VADER (github.com/cjhutto/vaderSentiment) is a rule-based sentiment analysis approach specifically sensitive to presumptions expressed on social media. It uses pre-annotated lexicons that are labeled as positive, negative, or neutral and categorize any sentence based on its lexicons' sentiments. VADER not only tells us about polarity but also provides information on how much positivity, negativity, and neutrality there is in a sentence. VADER has some limitations, but this approach is still considered a good option for

sentiment analysis from social media. Before applying VADER, data was pre-processed, and the first seven steps in Table 4 were adopted. The punctuations and stopwords were not removed from the data because VADER does consider such elements in the text. It is case-sensitive and gives different sentiment polarity for "english" and "English." VADER provides a probability distribution of sentiment for a sentence. It tells us the probability of a positive, negative, or neutral sentence. The sum of this distribution is always one. In addition, it provides an overall polarity score known as the "compound" score, which was selected for deciding on a tweet's sentiment in this study. A threshold was set for the compound score such that if the score was greater or equal to 0.05, the tweet was labeled as positive; if the score was less than or equal to −0.05, it was labeled as negative; otherwise, it was considered as a neutral one. VADER was applied to all datasets mentioned above, and the results are discussed in detail.

## TextBlob

TextBlob (github.com/sloria/TextBlob) is a simple Python library that works on a lexicon-based strategy. Here, the individual sentiments of lexicons constructing a sentence determine the sentence's sentiment. It uses "nltk" and allows us to perform extensive textual analysis and operations, including sentiment analysis. While implementing this approach returns two factors: polarity and subjectivity. The polarity score lies between −1 and 1, where −1 shows the negativity in the sentiment of a sentence and 1 implies the positivity. The negation is used to reverse the polarity of a sentence. The amount of "personal opinion" and actual data in a text is measured by "subjectivity." If a sentence has a higher value of subjectivity, it focuses more on "personal opinion" rather than the facts in the text. TextBlob returns zero if the sentence has no such word with a polarity score in the "nltk" lexicons-dataset because it calculates a sentiment score based on the weighted average of all the words present in a sentence. For the most part, TextBlob performs well in simple situations. Before applying TextBlob, data was pre-processed. Unlike VADER, the punctuations and stopwords were also removed from the data because TextBlob does not consider these elements. A threshold was also set here for the polarity score such that if the score was greater than zero, the tweet was labeled as a positive tweet; if the score was less than zero, the same was labeled as negative, and it was considered as a neutral one if the polarity score stood zero. This approach was also implemented on all datasets, and the results are further discussed in detail.

## Flair embeddings

Flair (github.com/flairNLP/flair) is an open-sourced simple library whose framework is built on top of PyTorch, which itself is a state-of-the-art deep learning framework. The flair library comes with plenty of useful features. It contains valuable word embeddings like ELMo (*Joshi, Peters & Hopkins, 2018*), GloVe (*Pennington, Socher & Manning, 2014*), character embedding, *etc*. Models like Flair are usually known as embedding-based models. The "Flair Embeddings" are the central embedding provided by this library which works based on contextual string embeddings. These use the inner states of different pre-trained models and provide new word embeddings. Using internal rules of multiple character

models can ensure consideration of the word's context. Like the other two previous approaches, data were pre-processed, and the first seven steps mentioned in Table 4 were adopted. The punctuations and stopwords were kept in the text to ensure that the model understands the context of tweets. This model returned two things: the sentiment label of the sentence and the model's confidence in labeling the sentence. As the flair model is trained on IMDb reviews, it could return only two sentiments, *i.e.*, positive or negative. The third sentiment, "neutral," could not be analyzed using flair embedding.

## Bidirectional encoder representations from transformers (BERT)

NLP has been achieving marvelous success in developing entities that can understand how best to represent the data, especially the text data, and understand the hidden meanings and relationships between different portions of the data. In addition, the NLP community has contributed several practical components that can be freely utilized in models developed for various tasks in the domain of NLP. The release of the BERT model is one of the most recent accomplishments amongst such achievements. This achievement is the start of a new era for NLP. BERT is a stack full of pre-trained Transformer encoders. This model has two different sizes: BERT Base and BERT Large. The base version has 12 encoder layers, while the large version has 24 layers. The model also has large feed-forwarded networks (hidden units), 768 in the base version and 1,024 in the large version, and attention heads, 12 and 16 for the base and large versions, respectively. As input, a sequence of words is given to BERT. Each encoder layer uses self-attention, then sends the results through a feed-forward network before passing them on to the next encoder layer. The first token is provided with a special token, CLS, which stands for classification. Another special token, SEP, which stands for separator, is used and added to separate the different sentences in the text. Each position generates a vector whose size is known as the "hidden_size." This size is 786 in the base version and 1,024 in the large version. The generated vector is then passed as an input to any classifier we want. The BERT model is fine-tuned and used to classify the sentiment in the study under discussion. The details of the process of fine-tuning are as follows:

### Training dataset

The dataset for training the BERT model is open-sourced and publicly available on GitHub (*Jadeja, 2022*). It contains 200,000 tweets related to COVID-19 collected in 2 years, 2020 and 2021. The dataset is passed through a few steps of pre-processing. The hashtags, user handles, URLs, duplicates, and null records are removed from the data. It is a highly balanced dataset with an equal number of records from all classes. There are three sentiment classes: positive, negative, and neutral; amongst them, 66,700 tweets are positive, 66,660 are negative, and 66,640 are neutral. There is a total of seven columns in the data: timestamp of the tweet, tweet ID, username, the content of the tweet, cleaned content of the tweet, sentiment label, and sentiment score. In the sentiment score, "2" is assigned to the positive label, "1" to the neutral, and "0" to the negative sentiment. The dataset was split into a 70–30% ratio for training and testing. The training set contains 140,000 tweets (70%), and the testing set contains 60,000 (30%). Both sets are also balanced and have an

equal number of each class. In the training set, 46,690 tweets are positive, 46,662 tweets are negative, and 46,648 records are neutral, while in the testing set, there are 20,010 positive, 19,998 negatives, and 19,992 neutral tweets. The balanced data ensures that a model would not lead to misleading results and give equal importance to each class present in the dataset.

### Model architecture

First, the class "AutoTokenizer" and "TFBertModel" from Transformers were used to load the tokenizer and the BERT model from the Hugging Face library. In this study, a "bert-base-cased" (huggingface.co/bert-base-cased) tokenizer and model were used to process our data to shape it into a form required by the BERT. The "bert-base-cased" is a base version of the BERT model and is case-sensitive. In our case, the tokens' length was just around 150, so to be on the safer side, the maximum length for the tokenizer was set to 160. The special tokens were added, and the "truncation," "padding," and "attention_mask" were set to True. As a result, the "input_ids" and "attention_mask" of the modeled data were returned. While building our model, we provided two input layers for the BERT layer. After that, the embeddings have hidden states. Then, using "GlobalMaxPooling1D" to build CNN layers, a dense layer of 128 units with "Relu" as the activation function, a dropout layer to avoid overfitting, a second dense layer with 32 units, and again "Relu" as the activation function, and at last, a dense layer with "3" units because there were three classes in our data with a "Sigmoid" function which led to the output of our function.

### Model training

For the compilation of the model built, Adam optimizer was used. The "learning_rate" was set to 5e-5 with a decay of 0.01, which is a bit slower but works perfectly. Since we were analyzing the multiclass problem, the loss was calculated from the "CategoricalCrossEntropy," and the balanced accuracy was taken from "CategoricalAccuracy" to include the average accuracy of all the classes in the data. First, the model was trained on ten epochs with a batch size of 16. The model was generalized well on data after ten epochs, but we trained on ten more epochs with the same batch size to achieve even better performance. Fig. 2 represents the training history of twenty epochs.

### Model evaluation

The model showed a validation accuracy of 89.02% after the first 10 epochs, which increased to almost 92% after the further ten epochs. The validation loss calculated during these phases was 0.3093 and 0.2315, respectively. After 20 epochs, the model had been generalized ideally on the data. The trained model was applied to all datasets discussed in the same section, and classification was performed. The results obtained are discussed from all angles in detail in the next section.

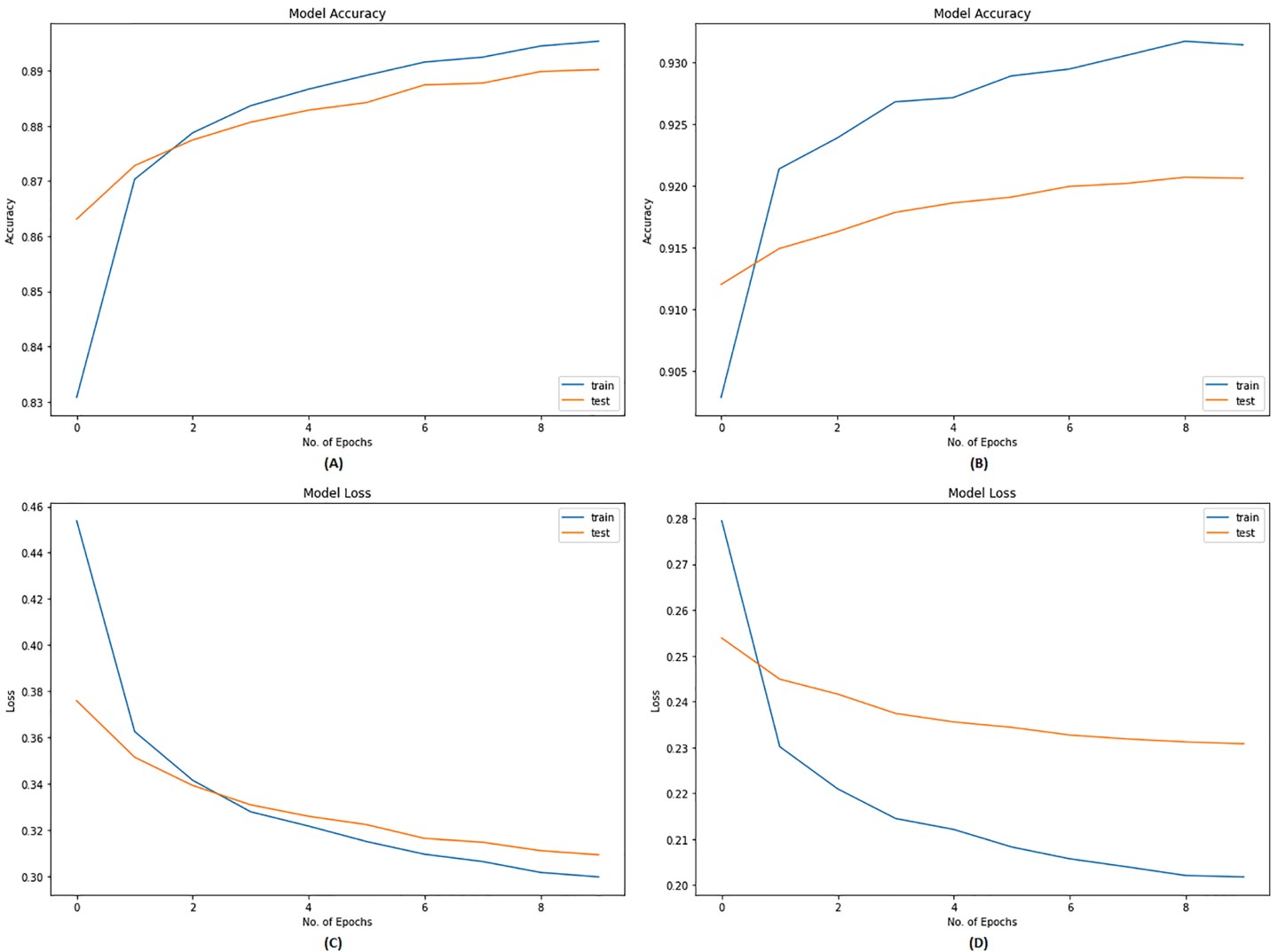

**Figure 2 Training history of BERT model.** (A) Accuracy graph after training on first 10 epochs. (B) Accuracy graph after training on next 10 epochs. (C) Loss graph after training on first 10 epochs. (D) Loss graph after training on next 10 epochs.

## RESULTS

Each approach showed its independent classification results on each dataset, whose illustrations and valuable insights are presented in this section. Table 5 represents the sentiments' distributions classified by all four models.

### Closure of educational institutes

Figure 3 depicts the sentiment percentages distribution per month about the closure of educational institutes. It can be observed from the classification results of VADER that the positive sentiment has a leading presence overall in the data than both the negative and neutral sentiments. In the year 2020, December, June, and April are the months with the highest percentage of positive, negative, and neutral sentiments, respectively, whereas, in

**Table 5 Sentiment percentage distribution done by all models.**

| Preventive measures | Model implemented | Sentiment percentages distribution | | |
|---|---|---|---|---|
| | | Positive | Negative | Neutral |
| Closure of educational institutes | VADER | 43.66% | 24.19% | 32.15% |
| | TextBlob | 39.36% | 16.26% | 44.37% |
| | Flair | 48.23% | 51.77% | N/A |
| | BERT | 33.55% | 18.98% | 47.47% |
| Suspension of flight operations | VADER | 41.30% | 31.26% | 27.44% |
| | TextBlob | 38.93% | 18.13% | 42.93% |
| | Flair | 38.48% | 61.52% | N/A |
| | BERT | 34.41% | 28.43% | 37.16% |
| Lockdown and closure of all business activities | VADER | 45.36% | 24.79% | 29.85% |
| | TextBlob | 38.93% | 18.13% | 42.93% |
| | Flair | 43.85% | 56.15% | N/A |
| | BERT | 36.36% | 20.57% | 43.07% |
| Enforcement of standard operating procedures | VADER | 46.18% | 27.95% | 25.87% |
| | TextBlob | 46.58% | 16.71% | 36.71% |
| | Flair | 34.88% | 65.12% | N/A |
| | BERT | 39.02% | 24.28% | 36.70% |
| Commencement of vaccination program | VADER | 44.31% | 22.42% | 33.27% |
| | TextBlob | 39.14% | 16.13% | 44.72% |
| | Flair | 32.33% | 67.67% | N/A |
| | BERT | 39.54% | 19.36% | 41.10% |

the year 2021, the months of February, May, and June contained the highest percentage of the same sentiments. Amongst both years, the positive sentiment was highest in February 2021. May 2021 was the month with the highest negative sentiment, while the neutral percentage was at its highest in April 2020. In the data classified by TextBlob, it can be noticed that the neutral sentiment has a leading presence overall in the data than positive and negative. In the year 2020, December, November, and March are the months with the highest percentage of neutral, positive, and negative sentiment, respectively, whereas, in the year 2021, the months of January, March, and May contain the highest percentage of the same sentiments. In both years, the neutral sentiment was highest in December 2020. November 2020 was the month with the highest positive sentiment, while the negative percentage was at its highest in May 2021.

The Flair model is trained on the IMDB reviews, and since there was no neutral sentiment, only two classes were considered here. We can see that the negative sentiment has a higher presence in the data than the positive. In 2020, October and March were the months with the highest positive and negative sentiment percentages, respectively. In the year 2021, the months of February and April contain the highest percentage of the same sentiments. In both years, positive sentiment was highest in February 2021, while the negative percentage was at its highest in March 2020. It can be observed from the

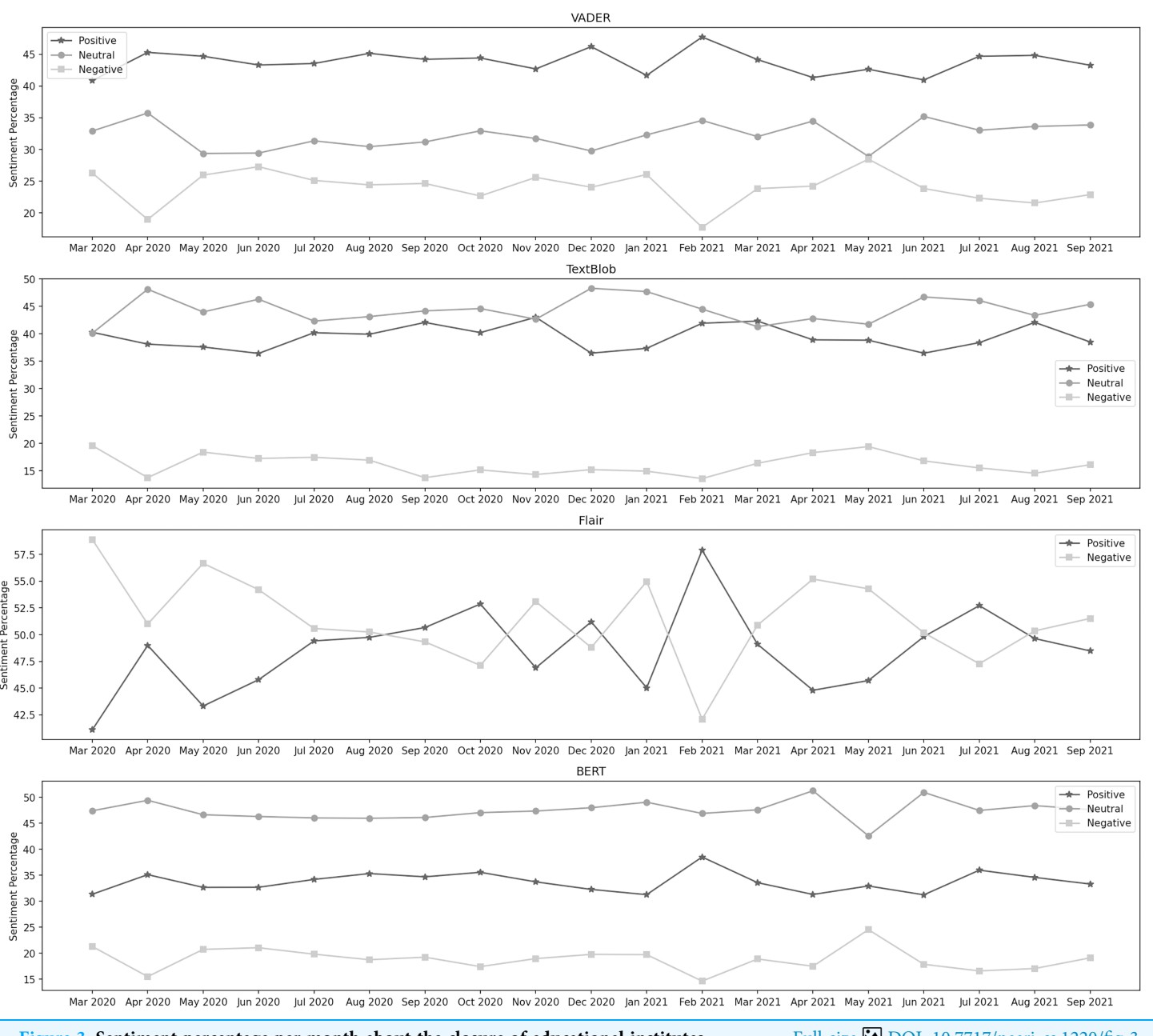

**Figure 3 Sentiment percentage per month about the closure of educational institutes.**

classification results of BERT that the neutral sentiment has a leading presence overall in the data than positive and negative. In the year 2020, April, October, and March are the months with the highest percentage of neutral, positive, and negative sentiment, respectively, whereas, in the year 2021, the months of April, February, and May contain the highest percentage of the same sentiments. During both years, the neutral sentiment was highest in April 2021. February 2021 was the month with the highest positive sentiment, while the negative percentage was at its highest in May 2021.

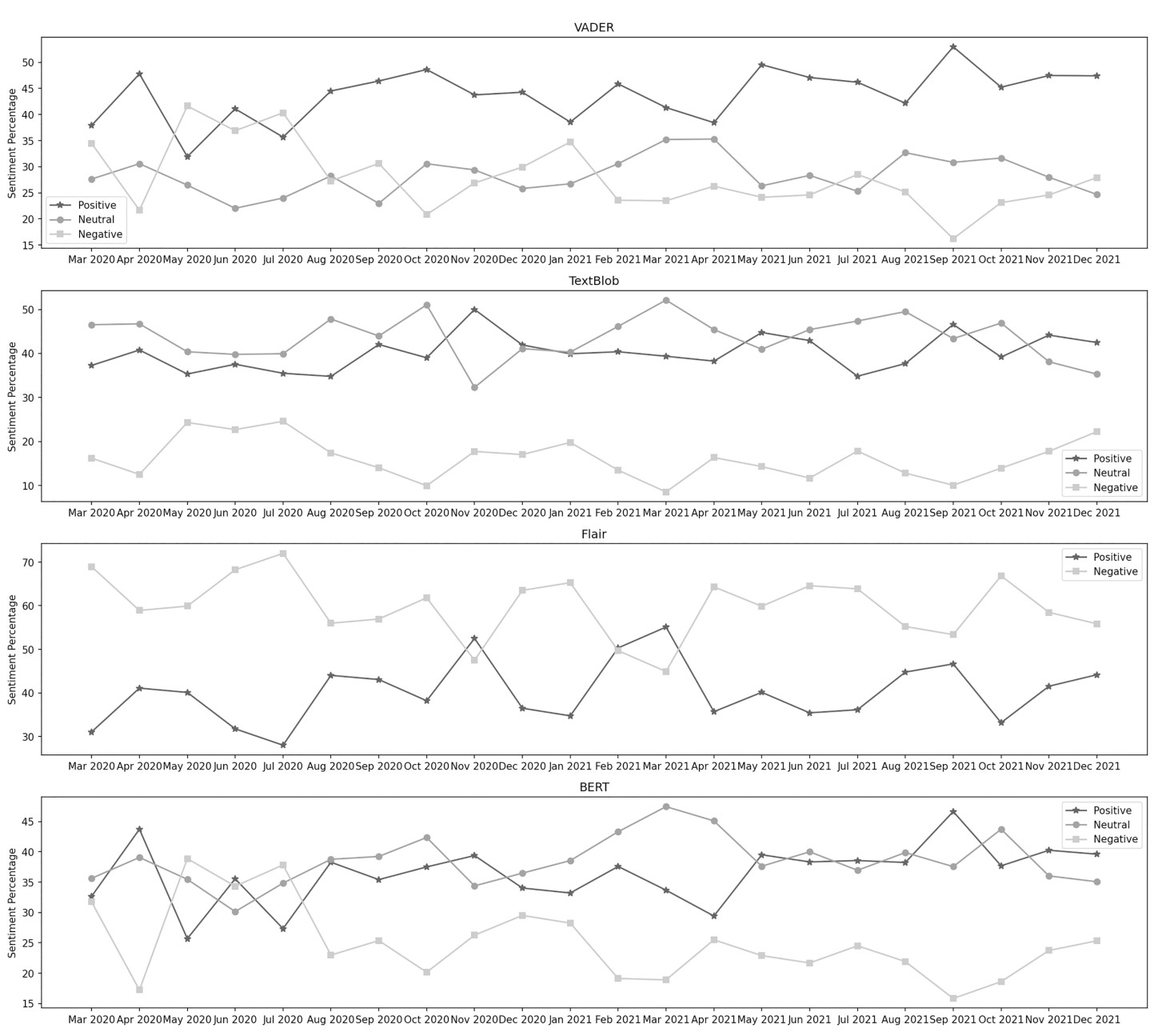

**Figure 4 Sentiment percentage per month about the suspension of flight operations.**

## Suspension of flight operations

Figure 4 depicts the sentiment percentages distribution per month about the suspension of flight operations. It can be observed from the classification results of VADER that the positive sentiment has a leading presence overall in the data than negative and neutral. In the year 2020, October, May, and April are the months with the highest percentage of positive, negative, and neutral sentiments, respectively, whereas, in the year 2021, the months of September, January, and April contain the highest percentage of the same

sentiments. Among both years, positive sentiment was highest in September 2021. April 2021 was the month with the highest neutral sentiment, while the negative percentage was at its highest in May 2020. In the data classified by TextBlob, the neutral sentiment has a higher overall presence than positive and negative. In the year 2020, October, November, and July are the months with the highest percentage of neutral, positive, and negative sentiment, respectively, whereas, in the year 2021, the months of March, September, and December contain the highest percentage of the same sentiments. During both years, the neutral sentiment was highest in March 2021. November 2020 was the month with the highest positive sentiment, while the negative percentage was at its highest in July 2021.

The results from the Flair model show that the negative sentiment has a higher presence in the data than the positive. In 2020, November and July were the months with the highest percentage of positive and negative sentiments, respectively. In 2021, March and October contained the highest percentage of the same sentiments. In both years, positive sentiment was highest in March 2021, while the negative percentage was at its highest in Jul 2020. It can be noticed from the classification results of BERT that the neutral sentiment has a leading presence overall in the data than positive and negative. In the year 2020, October, April, and May are the months with the highest percentage of neutral, positive, and negative sentiment, respectively, whereas, in the year 2021, the months of March, September, and January contain the highest percentage of the same sentiments. During both years, the neutral sentiment was highest in March 2021. September 2021 was the month with the highest positive sentiment, while the negative percentage was at its highest in May 2020.

## Lockdown and closure of all business activities

Figure 5 depicts the sentiment percentages distribution per month about the lockdown and closure of business activities. In the data classified by VADER, it can be observed that the positive sentiment has a leading presence overall in the data than negative and neutral. In the year 2020, December, June, and October are the months with the highest percentage of positive, negative, and neutral sentiments, respectively, whereas, in the year 2021, the months of November, May, and October contain the highest percentage of the same sentiments. Among both years, positive sentiment was highest in November 2021. October 2021 was the month with the highest neutral sentiment, while the negative percentage was at its highest in June 2020. The results from TextBlob show that neutral sentiment has a higher presence in the data than positive and negative. In the year 2020, March, December, and June are the months with the highest percentage of neutral, positive, and negative sentiments, respectively, whereas, in the year 2021, the months of October, November, and April contain the highest percentage of the same sentiments. During both years, the neutral sentiment was highest in October 2021. November 2021 was the month with the highest positive sentiment, while the negative percentage was at its highest in June 2020.

From the classification results of Flair, we can observe that the negative sentiment has a more leading presence overall in the data than the positive. In 2020, December and March were the months with the highest positive and negative sentiment percentages, respectively. In 2021, November and April contained the highest percentage of the same

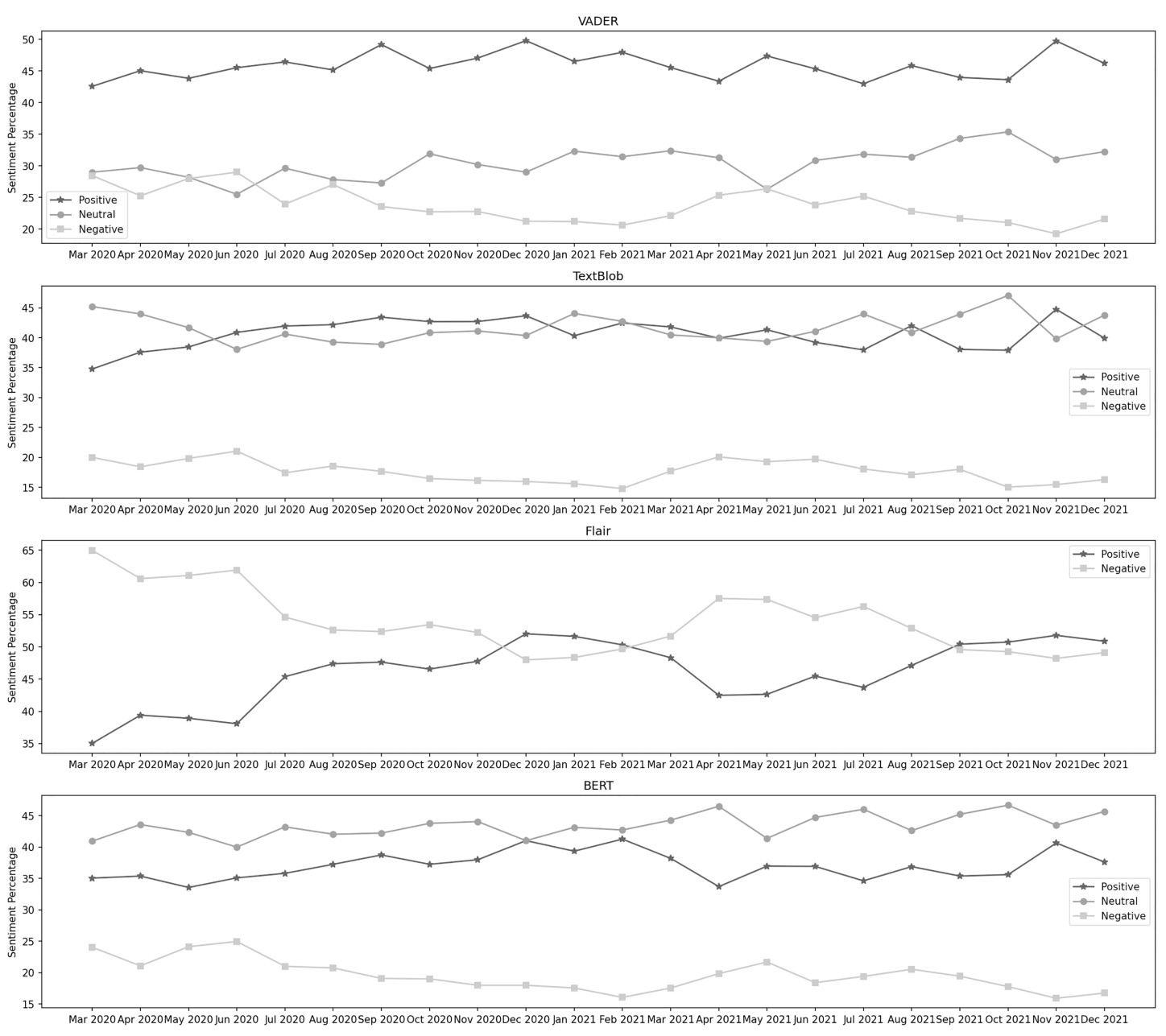

**Figure 5 Sentiment percentage per month about the lockdown and closure of business activities.**

sentiments. In both years, positive sentiment was highest in December 2020, while negative percentage was highest in March 2020. In the data classified by BERT, it can be noticed that the neutral sentiment has a leading presence overall in the data than positive and negative. In 2020, November, December, and June were the months with the highest neutral, positive, and negative sentiment percentages. In 2021, October, February, and May contained the highest percentage of the same sentiments. During both years, the neutral

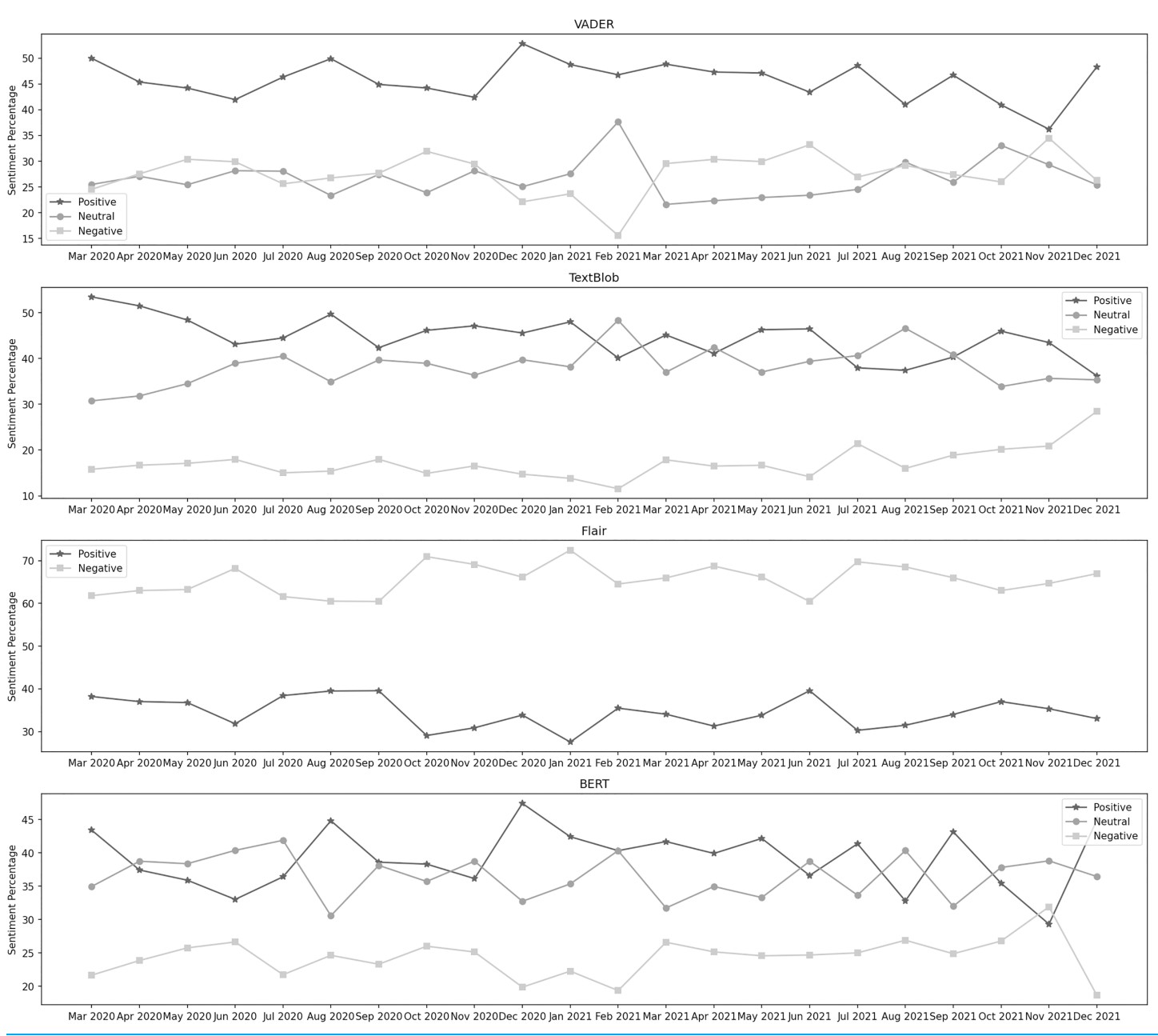

**Figure 6 Sentiment percentage per month about the enforcement of standard operating procedures.**

sentiment was highest in October 2021. February 2021 was the month with the highest positive sentiment, while the negative percentage was at its highest in June 2020.

## Standard operating procedures (SOPs)

Figure 6 depicts the sentiment percentages distribution per month regarding the enforcement of standard operating procedures. From VADER's results, we can see that the positive sentiment has a leading presence overall in the data than negative and neutral. In

the year 2020, December, June, and October are the months with the highest percentage of positive, negative, and neutral sentiments, respectively, whereas, in the year 2021, the months of March, February, and November contain the highest percentage of the same sentiments. Among both years, positive sentiment was highest in December 2020. February 2021 was the month with the highest neutral sentiment, while the negative percentage was at its highest in November 2021. The distribution by TextBlob shows that the positive sentiment has a leading presence overall in the data than neutral and negative. In the year 2020, March, September, and June are the months with the highest percentage of positive, negative, and neutral sentiments, respectively, whereas, in the year 2021, the months of January, December, and February contain the highest percentage of the same sentiments. Among both years, positive sentiment was highest in March 2020. February 2021 was the month with the highest neutral sentiment, while the negative percentage was at its highest in December 2021.

It can be observed from the classification results of Flair that the negative sentiment has a leading presence overall in the data than the positive. In 2020, September and October were the months with the highest positive and negative sentiment percentages, respectively. In 2021, June and January contained the highest percentage of the same sentiments. In both years, positive sentiment was highest in June 2021, while the negative percentage was at its highest in January 2021. BERT's results show that the positive sentiment has a higher presence in the data than neutral and negative. In the year 2020, July, December, and June are the months with the highest percentage of neutral, positive, and negative sentiment, respectively, whereas, in the year 2021, the months of August, December, and November contain the highest percentage of the same sentiments. During both years, the neutral sentiment was highest in July 2020. December 2020 was the month with the highest positive sentiment, while the negative percentage was at its highest in November 2021.

## Vaccination program

Figure 7 depicts the sentiment percentages distribution per month about the vaccination program. In the data classified by VADER, the positive sentiment has a higher overall presence than negative and neutral. In the first seven months, from December 2020 to June 2021, December, January, and February are the months with the highest percentage of positive, negative, and neutral sentiment, respectively, whereas, in the next 8 months, from July 2021 to February 2022, the months of January, November, and December contain the highest percentage of the same sentiments. Amongst all these months, positive sentiment was highest in January 2022. December 2021 was the month with the highest neutral sentiment, while the negative percentage was at its highest in January 2021. It can be observed from the classification results of TextBlob that the neutral sentiment has a leading presence overall in the data than positive and negative. In the first 7 months, from December 2020 to June 2021, January, March, and December are the months with the highest percentage of positive, negative, and neutral sentiment, respectively, whereas, in the next 8 months, from July 2021 to February 2022, the months of January, September, and December contain the highest percentage of the same sentiments. Amongst all these

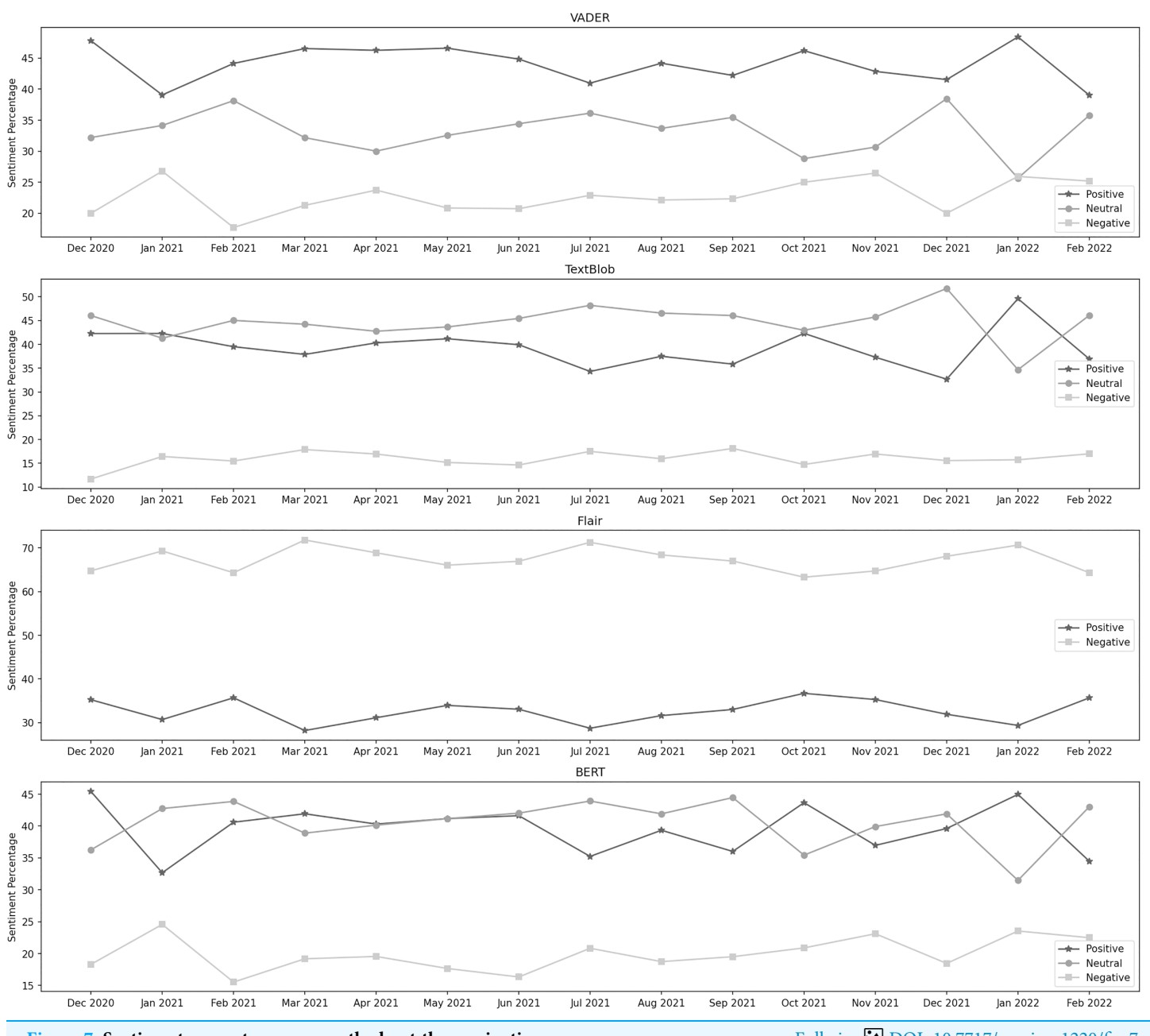

**Figure 7 Sentiment percentage per month about the vaccination program.**

months, positive sentiment was highest in January 2022. December 2021 was the month with the highest neutral sentiment, while the negative percentage was at its highest in September 2021.

The results from the Flair model show that the negative sentiment has a leading presence overall in the data than positive. In the first 7 months, from December 2020 to June 2021, February and March are the months with the highest percentage of positive and negative sentiment, respectively, whereas, in the next 8 months, from July 2021 to February

2020, the months of October, and July contain the highest percentage of the same sentiments. Amongst all these months, positive sentiment was highest in October 2021, while the negative percentage was at its highest in March 2021. We can notice from BERT's results that the neutral sentiment has a leading presence overall in the data than positive and negative. In the first 7 months, from December 2020 to June 2021, December, January, and February are the months with the highest percentage of positive, negative, and neutral sentiment, respectively, whereas, in the next 8 months, from July 2021 to February 2022, the months of January, January, and September contain the highest percentage of the same sentiments. Among all these months, positive sentiment was highest in December 2020. September 2021 was the month with the highest neutral sentiment, while the negative percentage was at its highest in January 2021.

## Summary

The implemented approaches are based on different techniques, which have been discussed earlier in detail. Each of them has its capabilities and limitations. VADER is considered a powerful tool for social media analysis; the lexicon vocabulary used by VADER is not updated regularly, and it is challenging for this approach to evaluate the slang of this modern era, which is used very extensively in social media and change day by day. TextBlob is also a rule-based technique that calculates the polarity score of individual terms and determines the overall sentiment of the sentence. Flair embeddings are trained on IMDb reviews. The sentimental tone of a review and a social media post is usually very different, and such pre-trained embeddings cannot be accurately used for the sentimental analysis of a social media platform toward a specific topic. Another crucial aspect to focus on in sentiment analysis is the context of a sentence. The context must be preserved while determining the sentimental attitude of the text. Considering all these factors, building a custom model and fine-tuning it for a specific purpose becomes essential. A fine-tuned model can overcome such drawbacks and give the most accurate results. In the study under discussion, a fine-tuned BERT model is used, which is discussed in the previous section in detail. As this model is trained on a labeled Twitter dataset and contains all the capabilities of a state-of-the-art transformers-based model, the results shown by this custom-trained model were considered the most accurate and taken to a noticeable conclusion. These results were analyzed and compared at the time of four COVID-19 waves in the country during the years 2020 and 2021.

By the end of 2021, Pakistan faced four dangerous coronavirus waves. The first wave was noticed from March 2020 till the end of July 2020, the second wave remained from the start of November 2020 till the end of January 2021, the third wave lasted from March 2021 till the end of May 2021, and the fourth and last wave in these 2 years was observed from July 2021 to September 2021. The preventive measures under discussion were implemented across the country in different phases during these four waves. The time-to-time public sentiment about these measures is discussed as follows:

1. All educational institutes, including schools, colleges, universities, and madrasas, were closed across the country. The examinations got postponed for the time being. Later,

most institutes, excluding primary and middle schools, were allowed to resume their activities online. After a period, fifty percent of strength got permitted to attend face-to-face classes. These developments continued to occur in various situations depending on the pandemic scenario in the country (*The Express Tribune, 2020b*; *The International News, 2020*; *Ary News, 2020b*). The overall sentiment remained neutral in the data about the closure of educational institutes. During the first wave, in March 2020, almost 48% of the sentiment distribution was neutral, while the positive and negative distribution was 31% and 21%, respectively. In April 2020, the negative sentiment decreased to 16%, while the positive and neutral percentages increased to 35% and 49%, respectively. In May 2020, there was an increase in the negative sentiment, whereas both other percentages got slightly decreased. This distribution level remained almost the same for the next 2 months. At the start of the second wave, in November 2020, the positive, neutral, and negative sentiment percentage was 34%, 47%, and 19%, respectively. The positive sentiment decreased in the next 2 months, while neutral and negative sentiments increased. At the end of the second wave, the percentage of the same sentiments was 31%, 49%, and 20%, respectively. During the third wave, in March 2021, 48% of the sentiment was neutral, while 33% and 19% were positive and negative, respectively. In the next month, a significant increase was observed in the neutral sentiment. In the third month, there was a massive decrease in the neutral percentage, whereas the negative sentiment increased. At the time of the last wave, in July 2021, the neutral sentiment percentage was 47% which went up to one point in August 2021 and again came down to 47% in the month of September 2021. The positive sentiment was noticed at 36% in July 2021. It gradually decreased in the next 2 months and was found to be 33% in the last month of the fourth wave. The negative sentiment followed the opposite way and increased from 17% to 19% from the first to the last month, September 2021. Overall, public behavior remained neutral or positive about the closure of educational institutes throughout the period.

2. Following the other measures, all international and domestic flight operations got suspended. However, the officials arranged a few special flights to bring some nationals stuck overseas. These protocols kept changing following the policies set by the countries on the other side of the routes (*The Guardian, 2020*; *Kaleej Times, 2020*; *Business Today, 2020*). The public sentiment about the suspension of flight operations kept changing on a significant scale throughout the period. During the first wave, in March 2020, it was as follows: 35% neutral, 33% positive, and 32% negative. In April 2020, there was a massive hike in positive sentiment, which increased to 44%, and the neutral sentiment also increased to 39%. A substantial decrease was observed in the negative sentiment. In May 2020, the positive and neutral sentiments decreased to 26% and 35%, respectively, whereas the negative sentiment increased to 39%. In June 2020, the neutral distribution decreased to 30%. The negative distribution also got down to 34%, while the positive sentiment increased to 36%. In July 2020, the neutral and negative sentiments increased to 35% and 38%, respectively, and the positive percentage went down. In the second wave, in November 2020, the positive, negative, and neutral sentiment distributions

were 39%, 26%, and 35%, respectively. In December 2020, the same distribution went to 34%, 30%, and 36%. In the last month of the second wave, the neutral distribution slightly increased, whereas both other sentiments observed a slight decrease. During the third wave, in March 2021, the positive sentiment percentage was 34%, while the neutral and negative percentages were 47% and 19%, respectively. Next month, the negative percentage increased to 26%. In May 2021, a significant increase was noticed in positive sentiment. At the time of the last wave, in July 2021, the positive sentiment was 39% of the overall distribution, which went to 38% in August 2021 and increased to 47% in September 2021. The neutral distribution remained at 37%, 40%, and 37% in July, August, and September 2021, respectively. The neutral sentiment was 24% in July 2021 and gradually decreased in the next 2 months by the end of the fourth wave.

3. When the number of cases was getting higher daily, the federal and provincial governments imposed a strict lockdown at different levels. All public areas were closed, excluding pharmacies, grocery shops, and vegetable stores. The offices, wedding halls, parks, hotels, restaurants, and eateries were closed. The govt. also banned interprovince and intercity transport at different times. The country later opted for a smart-lockdown strategy; only hot areas were cordoned off wherever required (*The Express Tribune, 2020a*; *Arain, 2020*; *Ary News, 2020a*; *OyeYeah, 2020*; *Statesman, 2020*). It can be observed that the overall sentiment about the lockdown of all business activities remained neutral in both years. During the first wave, in March 2020, almost 41% of the sentiment distribution was neutral, while the positive and negative distribution was 35% and 24%, respectively. In April 2020, the negative sentiment decreased to 21%, while the neutral percentage increased to 44%. In May 2020, there was a decrease in positive and neutral sentiments. At the start of the second wave, in November 2020, the percentage of positive, neutral, and negative sentiment was 38%, 44%, and 18%, respectively. Next month, the positive percentage increased to 41%, while the neutral percentage decreased. In January 2021, a slight decrease in positive sentiment was observed, whereas the neutral sentiment increased slightly. During the third wave, in March 2021, 44% of the sentiment was neutral, while 38% and 18% were positive and negative, respectively. In April 2021, a significant decrease was observed in the positive sentiment. In May 2021, there was an increase in the positive and negative percentages, which went to 37% and 22%, respectively. At the time of the last wave, in July 2021, the neutral sentiment percentage was 46% which went down to 43% in August 2021 and again increased to 45% in September 2021. The positive sentiment was noticed at 35% in July 2021. It increased in the next month to 37% and was found to be 35% in the last month of the fourth wave. The negative sentiment slightly went up from 19% to 20% from July 2021 to August 2021 and remained the same in September 2021. The public sentiment remained neutral or positive about the lockdown of all business activities.

4. People were asked to comply with multiple SOPs set by WHO, including mask-wearing, hand sanitization, social distancing, *etc*. Several restrictions were also imposed against the non-compliant people (*Al-Jazeera, 2020*; *Dawn, 2020*; *Ahmed, 2020*; *Geo News, 2020a*; *Associated Press of Pakistan, 2020*). The public sentiment about the SOPs

remained positive or neutral throughout the period. During the first wave in March 2020, the sentiment distribution was 35% neutral, 43% positive, and 22% negative. A decrease in positive sentiment went to 37% in April. During May 2020, there was a slight decrease in positive and neutral sentiments. In June 2020, the positive distribution decreased to 35%. The neutral and negative percentages increased to 40% and 27%, respectively. In the last month of the first wave, the positive and neutral sentiments increased to 36% and 42%, respectively, and the negative percentage went significantly down to 22%. During the second wave, in November 2020, the positive, negative, and neutral sentiment distributions were 36%, 39%, and 25%, respectively. In December 2020, the same distribution went to 47%, 33%, and 20%. In the last month of the second wave, the positive distribution decreased, whereas both other sentiments observed an increase. During the third wave, in March 2021, the positive sentiment percentage was 42%, while the neutral and negative percentages were 32% and 26%, respectively. Next month, the neutral percentage increased to 35%, while both other sentiments' distributions decreased slightly. In May 2021, an increase was noticed in positive sentiment. The neutral and negative distributions reduced a bit. At the time of the fourth and last wave, in July 2021, the positive sentiment was 41% of the overall distribution, which went to 33% in August 2021 and increased to 43% in September 2021. The neutral distribution remained at 34%, 40%, and 32% in July, August, and September 2021, respectively. The neutral sentiment was 25% in July 2021; it increased to 27% in the next month, August 2021, and went again to 25% in September 2021 by the end of the fourth wave.

5. The initiation of the mass vaccination program was a significant relief for the public across the country. First of all, medical and paramedical staff was vaccinated. It was made available for the people, categorizing them with their age groups, and mandatory for every citizen above ten later (*Geo News, 2020b*; *Geo News, 2021*; *Dawn, 2021*; *Al-Jazeera, 2021*; *The Express Tribune, 2021*). December 2020 and January 2021 were analyzed to test the sentiment about the COVID vaccine before it was publicly available. It can be observed that the sentiment remained positive and neutral throughout the period. In December 2020, the positive sentiment percentage was 46%, while the neutral and negative sentiment distributions were 36% and 18%, respectively. The positive sentiment decreased to 33% in the next month, while the neutral sentiment went up to 43%. Pakistan started its COVID-19 vaccination drive on February 02, 2021. It was first made available for the front-line medical and paramedical staff and later for the public. In February 2021, the sentiment distribution was as follows: positive 41%, neutral 44%, and negative 15%. The vaccine was made available to people over 60 on March 10, 2021. In March 2021, the positive sentiment slightly increased to 42%. People over 50 were allowed to get vaccinated from March 30, 2021. In April 2021, the percentage distribution was as follows: positive 40%, neutral 40%, and negative 20%. From May 03, 2021, the age limit was 40 for vaccination eligibility. In May 2021, the positive and neutral sentiments slightly increased to 41% each. The Government of Pakistan allowed people over 30 to get their vaccination shots from May 30, 2021, and after a few days,

from June 03, 2021, all over 18 years of age were allowed to get vaccinated. In June 2021, the sentiments were as follows: positive 42%, neutral 42%, and negative 16%. Next month, the positive sentiment got decreased to 35%. In August, the positive sentiment again increased and went to 39%. The neutral and negative percentages got slightly decreased. August 30, and 31, 2021, were the days when the highest number of vaccination doses were administrated to the public. Over the next 6 months, the positive and neutral sentiments remained dominant over the distribution.

## DISCUSSION

All the implemented approaches were performed independently, and the results obtained from the state-of-the-art BERT model were considered the most accurate because of its capabilities to overcome the limitations of the other three approaches. Five preventive measures were selected for the analysis, including the closure of educational institutes, suspension of domestic and international flight operations, lockdown of all business activities, enforcement of standard operating procedures, and the immunization drive started by the Government of Pakistan. The public sentiment during the four waves of the coronavirus was analyzed, and it came under observation that the sentiments were neutral throughout the period about the closure of educational institutes. The individual distribution kept varying from month to month, but overall, it remained neutral, followed by an influx of positive sentiments. During the first wave, people's sentiments were inconsistent about the suspension of flight operations. Initially, it was positive, but the negative distribution spiked the next month. However, it kept switching between positive and neutral sentiments during the subsequent waves. The sentiments about the lockdown and closure of all business activities were noticed as neutral. December 2020 was the month during which the positive and neutral distributions were almost equal; otherwise, the percentage remained neutral. It was observed that the people also remained positive or neutral about enforcing standard operating procedures (SOPs) too. Only in November 2021 the negative distribution was above the positive one; however, this month's neutral sentiment was on top. Pakistani people showed the same sentiment about the vaccination program also. It remained positive or neutral during the 15 months considered for the analysis. Pakistan had never faced an emergency on such a large scale before the emergence of COVID-19. The countermeasures deployed by the governing authorities were nothing less than setting a new course of daily life for every individual. The measures analyzed in this study are the ones that have affected public life the most. The closure of educational institutes for such a long period and then shifting the majority of them to an online mode of action was not routine for the general public. The suspension of flight operations was also affecting the whole community. Many overseas Pakistanis and people working abroad got stuck in their respective countries, and there was no way to bring them back to Pakistan. Imposing the lockdown, which was shifted to a smart-lockdown strategy later, and fixing the business hours impacted the citizens extensively. Many SOPs, including wearing face masks and gloves, hands-sanitization, maintaining social distance, and others, were the things the public was facing for the first time. When COVID vaccines were in the

laboratory phase and not available to the community, there was a mixed reaction in public about them. Later, even during the vaccination campaign at the masses level, they faced vaccination hesitancy in some groups of people. As mentioned earlier, these were the steps the community met for the first time in life. The people were dying, and no one could predict anything about the pandemic. It can be expected public sentiments to vary at different levels during such a scenario. Based on the observations, we can say that the public showed appreciative, either positive or neutral, behavior towards the preventive measures taken by the Government of Pakistan over time containing the spread of the deadly coronavirus.

## CONCLUSION

The research under discussion is performed to conduct a comprehensive analysis to observe the public sentiment about the countermeasures taken by the Government of Pakistan in combating the COVID-19 disease. Such analyses are considered especially important because the response from the public is a crucial factor when implementing state-level policies. Data from the microblogging social platform Twitter was analyzed during the years 2020 and 2021. Five comprehensive datasets about different preventive measures are collected, which include the closure of educational institutes, suspension of flight operations, lockdown of business activities, enforcement of several standard operating procedures (SOPs), and the commencement of the vaccination program. Four approaches, VADER, TextBlob, Flair, and BERT, are implemented for the analysis. The first two, VADER and TextBlob, are rule-based techniques, while Flair and BERT are automated or learning-based approaches. BERT is fine-tuned on pre-labeled data, which showed a validation accuracy of 92% after 20 epochs. The public sentiment at the time of four COVID-19 waves during the years 2020 and 2021 is compared, and it is noticed from the results of the state-of-the-art transformer-based BERT model that the people of Pakistan had either neutral or positive sentiments about the counter strategies implemented against the spread of the coronavirus. Although this study is a comprehensive one and answers many of the sought questions, it still has some limitations which can be considered for improvements while performing some related work in the future:

- This study is performed on English-language tweets, while most Pakistani people speak Urdu, the country's national language. A similar sort of analysis can be done on the data in the Urdu language, which can present the picture in a bigger frame.
- Some important factors such as emojis, emoticons, and emphasized words, like tooooo good, contribute to determining the degree of the sentiment of a text, especially on social media. They help us know about the sentiment level of a particular piece of text. A study can be proposed in the future, probing the effects of including and excluding such factors.
- The sentiment may have several reasons to be positive, neutral, or negative. To get much better insights, topic modeling, a process of determining the semantic structures from text, and aspect-based sentiment analysis, which is a text analysis method that divides

opinions into aspects and determines the sentiment associated with each, can be performed on all datasets discussed in this study.

- Sarcasm is an aspect that can affect sentiment analysis in various ways. A generic sentiment analysis study cannot consider sarcasm as well. A study can be carried out on detecting sarcasm in the discussed data.
- An aspect-based sarcasm detection can be a comprehensive and contributing angle in this scenario.

### Funding
The authors received no funding for this work.

### Competing Interests
The authors declare that they have no competing interests.

### Author Contributions
- Muhammad Faisal Ali conceived and designed the experiments, performed the experiments, analyzed the data, performed the computation work, prepared figures and/or tables, authored or reviewed drafts of the article, and approved the final draft.
- Rabia Irfan conceived and designed the experiments, performed the experiments, analyzed the data, performed the computation work, prepared figures and/or tables, authored or reviewed drafts of the article, and approved the final draft.
- Tahira Anwar Lashari analyzed the data, authored or reviewed drafts of the article, also provided invaluable assistance with the reviews and comments given by the editor and reviewers, and approved the final draft.

### Data Availability
The raw data is available at Zenodo: mohdfaisalali. (2022). mohdfaisalali/public-sentiments-about-pakistan-s-anti-covid-measures: v.01 (v.01). Zenodo. https://doi.org/10.5281/zenodo.7360872.

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
