# Peer review of "Comprehensive sentimental analysis of tweets towards COVID-19 in Pakistan: a study on governmental preventive measures"

_PeerJ Computer Science, doi:10.7717/peerj-cs.1220_

## Round 0.1 · original submission · Major Revisions

English needs to be improved.

Reviewer 1 ·

Basic reporting

The English Language used in the article is poor. In terms of technical terms, it has reflected good LR.

I am having following suggestions:
1. the format of the paper is good but the text is not justified.
2. Overall good effort.

Experimental design

I am having following suggestions:
1. the format of the paper is good but the text is not justified.
2. I didnt see any data from Pakistan even in tables. I have a fear that the datasets are not appropriate if
you claim Pakistan in title then the tables & the plots must have Pakistan in it.
3. Overall good effort.

Validity of the findings

Serious concern as title represents Pakistan Whereas I cant see anything in dataset & plots.

Additional comments

Read my all comments

Reviewer 2 ·

Basic reporting

The paper describes four approaches analyzing sentiment from tweets dealing with COVID-19.
The authors also claim that five datasets, in this context, were built but they are not clearly introduced in the paper. They don’t mention if they are made available to the community. The document is organized and clear, and experimental results are more or less convincing. However, many points/sections need to be clarified.

It seems that there is a confusion between opinion mining and sentiment analysis! The sentiment is the feeling of the person writing the tweet while the opinion is the position of the person toward something. We can have a positive sentiment and negative opinion about something and vice versa.
I think that the authors should be definite and consistent.

Experimental design

It is not well justified why authors remove hashtags during preprocessing? Particularly for twitter, hashtags are a relevant source of information and can sometimes encode sentiments.

I would have appreciated if the authors exploit the particular linguistic phenomena on social networks in order to extract feelings like emojis, repeated letters (goood, tooo bad) ...

Validity of the findings

The article offers a quite rich and informative discussion. But I would like the authors to further discuss why sentiments regarding a preventive measure can change over time and what are exactly the decisions in relation with each measure that were appreciated by people?

Additional comments

Minor remark:
Please add as footnotes the links to the tools used in this work (Snscrape, vader, TextBlob...)

---

## Round 0.2 · accepted · Accept

The paper is now acceptable.

Reviewer 1 ·

Basic reporting

The authors have shown quality work & overall they have improved the manuscript.

Experimental design

The research is well defined & meaningful. The gaps were highlight & identified & solved very well.

Validity of the findings

All underlying data have been provided; they are robust, statistically sound, & controlled

Reviewer 2 ·

Basic reporting

The authors have made the necessary improvements and many choices have been justified.
For me, the paper can be accepted for publication.

Minor: the lines indicated in the rebuttal letter do not correspond to the modified lines in the pdf. This may be due to the fact that the paper undergoes layout changes before sending to proofreaders

Experimental design

nothing to add

Validity of the findings

nothing to add